

**Horizontal ridging with mulching as the optimal tillage practice to reduce surface**
**runoff and erosion in a Mollisol hillslope**
Running title: Horizontal ridge with mulching reduces surface runoff and erosion
Yucheng Wang[1, †], Dayong Guo[2, †], Zheng Li[1], Wuliang Shi[1], Bin Li[1], Liyuan Hou[1], Yi Zhang[1], Jinhu
Cui[1], Ning Cao[1, *], and Yubin Zhang[1, *]
[1] College of Plant Science, Jilin University, Changchun 130062, China
[2] Agricultural College, Henan University of Science and Technology, Luoyang 471000, China
†, contribution equally to this work
*, Correspondence
Ning Cao and Yubin Zhang, College of Plant Science, Jilin University, Changchun 130062, China.
E-mail: cao_ning@jlu.edu.cn; ybzhang@jlu.edu.cn
ORCID
Yucheng Wang, https://orcid.org/0000-0002-7838-226X
Wuliang Shi, https://orcid.org/0000-0002-8023-1807
Yubin Zhang, https://orcid.org/0000-0003-4920-3100



**ABSTRACT**
Soil erosion features and ideal tillage practices are not very clear at the crop seedling stage in Chinese
Mollisols. Simulated rainfall experiments were conducted at the rainfall intensities of 50 and 100 mm
h$^{-1}$ to investigate the differences in soil erosion of a 5° hillslope during the maize seedling stage
between conservation and conventional tillage measures, including cornstalk mulching (Cm),
horizontal ridging (Hr), horizontal ridging + mulching (Hr+Cm), vertical ridging + mulching
(Vr+Cm), flat-tillage (CK), and vertical ridging (Vr). The results demonstrated that crops could remit
soil erosion at the seedling stage by reducing the kinetic energy and changing the distribution of
raindrops. The conservation tillage measures significantly alleviated total runoff (11.7%–100%) and
sediment yield (71.1%–100%), postponed runoff-yielding time (85 s–26.1 min), decreased runoff
velocity (71.5%–96.7%), and reduced runoff and soil loss rate, compared to the conventional tillage
measures. Practices with mulching showed better performance than Hr. Mulching reduced sediment
concentration (~70.6%–100%) by decreasing runoff velocity and soil particle filtration in a manner
similar to buffer strips. The contour ridge ruptured earlier at 100 mm h$^{-1}$ than at 50 mm h$^{-1}$ and changed
the characteristics of the soil erosion by providing a larger sediment source to the surface flow. Runoff
strength, rather than soil erodibility, was the key factor affecting soil erosion. Decreasing runoff
velocity was more important than controlling runoff amount. The Hr + Cm treatment exhibited the
lowest soil erosion and is, thus, is recommended for adoption at the corn seedling stage in sloping
farmlands.
**KEYWORDS**
soil erosion, conservation tillage, Mollisols, maize seedling stage, rainfall simulation, rainfall
intensity





## Introduction

Soil erosion has been accelerated by unsustainable agricultural practices (FAO, 2019), with an
associated annual loss of $8 billion to the global GDP, global agri-food production by 33.7 million
tons, and 48 billion m$^3$ water (Sartori et al., 2019). Sloping farmlands are considered as the main sites
of soil erosion worldwide (Ge et al., 2021; Haddadchi et al., 2019). With the removal of fertile soil
surface layers following intensive tillage, soil erosion leads to soil layer thinning, soil quality
degradation, and crop yield decline (DeLonge and Stillerman, 2020; Liu et al., 2013).
Mollisols regions, which are found in flat to undulating land (Chesworth, 2008), are the major
crop production areas globally while experiencing severe soil erosion from the 1930s to date due to
overexploitation (Zheng, 2020). Expansive acres of maize (*Zea mays* L.) are grown on slopes (You et
al., 2021) due to the naturally fertile mollic epipedon and high productivity in the Mollisols of
Northeast China (Zhao et al., 2015), which account for 46.39% of the total soil loss area in the region
(MWR, 2020). Hence, addressing soil erosion is important for soil loss reduction, aquatic ecosystem
conservation, and agricultural sustainable development in the region.
Conservation tillage is one of the widely used agronomic measures worldwide to control soil
erosion (Bombino et al., 2021; Busari et al., 2015; Kader et al., 2017; Lal, 2018). Compared with
conventional tillage approaches, conservation tillage improves soil physical characteristics (Blanco-
Moure et al., 2012), soil fertility (Van den Putte et al., 2012), and agricultural productivity (Hansen et
al., 2012).
Few studies have explored the active influences of crops on soil erosion, especially at the seedling
stage (Cerdà et al., 2017; Prosdocimi et al., 2016b; Wang et al., 2018), although some previous studies
have demonstrated the significant and positive effects of vegetation on soil erosion (Huang et al., 2014;
Wang et al., 2021b). In addition, although some reports have explored the effects of conservation
tillage on soil erosion by simulating rainfall in the region, they have focused on bare slopes or have
been limited to the rainy season from July to September (Li et al., 2016; Liu et al., 2011; Lu et al.,



2016; Xu et al., 2018). The status of soil erosion at the crop seedling stage under different tillage
practices has rarely been explored (Ma et al., 2013). Sloping farmland is susceptible to soil erosion at
the seedling stage (Zhang et al., 2010) for various reasons, including low vegetation cover and poor
soil holding capacity (Figure 1) (Wang et al., 2018; Zhang et al., 2009) with the advance of
precipitation concentration period (Liu et al., 2018; Sun et al., 2000;).
The objectives of the present study were to 1) identify influence of maize seedling canopy on soil
loss and 2) evaluate the effects of four conservation tillage and two conventional tillage practices on
soil erosion under simulated rainfall conditions on a black soil sloping farmland. The results of the
present study could provide insights on the optimal tillage approaches at corn seedling stages in
Mollisol regions, which could facilitate soil erosion control measures in such regions.
**Materials and Methods**
**Study area and rainfall simulation**
The experiments were conducted at artificial rainfall simulation plots at the Science and
Technology Park of Soil and Water Conservation (127°25'35.8788"E, 45°45'22.3308"N), Institute of
Soil and Water Conservation of Heilongjiang Province, Binxian County, which belongs to the typical
Mollisol region, gentle (1-8°) and long slopes (~400-1000 m) are the key topographical features, in
Northeast China, the annual average precipitation is 548.5 mm and 64% of the precipitation
concentrated in summer (MWR, CAS, and CAE, 2010).
The rainfall simulation device adopted is composed of a water storage system, a control system,
and a sprinkler system (Wen et al., 2012). The sprinkler system is erected 6-m from the ground. A full-
jet down-sprinkler rainfall simulator (Spraying Systems Co., Wheaton, IL USA) with three nozzle
sizes (Fulljet 1/8, 2/8, and 3/8) was used to apply rainfall. Rainfall intensity can be adjusted from 20
to 150 mm h$^{-1}$. Wen et al. (2012) reported that the uniformity coefficient of rainfall intensities from 30
mm/h to 90 mm/h was ~0.90. The control system is a HLJSB-J artificial rainfall simulation system
(Institute of Soil and Water Conservation of Heilongjiang). A removable waterproof canvas ceiling



89 was used protect all experimental plots from natural rainfall, and a set of droppable canvases were

90 used to surround the testing plots to eliminate the impacts of wind (Figure 2 and 3).

91 **Preparation of experimental plots**

92  The plots used in the present study were 10 m long and 1 m wide. The slope of the plots was set

93 to 5° to simulate the typical natural geomorphological conditions in farmlands in the region (Zhao,

94 1986). The depth of the tested black soil was 0.3 m, similar to the average thickness of the A-horizon

95 of black soil in Binxian county (Xu et al., 2010). The black soil layer was followed by a 0.3-m sand

96 layer.

97  The used soil was Phaeozems (IUSS Working Group WRB. 2015), same as typical black soil

98 (CRGCST, 2001) or mollisol (Soil Survey Staff, 1999), with 22.01 g kg$^{-1}$ of organic matter and

99 approximately 7.9% sand, 54.4% silt, and 37.7% clay, determined using the potassium dichromate

100 oxidation-external heating method and density method with variable depth, respectively (Pansu and

101 Gautheyrou, 2005). The soil was collected from the top-30-cm soil layer in a local sloping farmland.

102 The agglomerate impurities were removed manually, but without passing the soil through a sieve, to

103 maintain its natural status. The soil was packed into plots on the sand layer for 1.5 years to ensure that

104 the bulk density (1.20 g cm$^{-3}$), determined by the core method (Lampurlanés and Cantero-Martínez,

105 2003; Liu, 1996; Soil Survey Staff, 2009), reached the field level by natural deposition, and the soil

106 structure recovered to the natural cropland state before the experiment.

107  We used Xianyu 335 maize variety (DuPont Pioneer Ltd., USA), a widely cultivated variety in

108 Northeast China (Liu et al., 2021). Seeds were sown with 0.4-m spacing between rows and 0.2-m

109 spacing between plants, and fertilized with urea ($CO_2(NH_2)_2$) at 150 kg ha$^{-1}$ on June 9, 2013. All plots,

110 excluding the flat-planting plots, were plowed simultaneously at ~0.2 m depth. Ridges, 15 cm high

111 and 15 cm wide, were stacked in all ridging plots one month after sowing based on the local methods

112 (Wang, 2015). Air seasoning maize stalks were chopped into approximately 5-cm fragments and

113 mulched onto mulching plots at a rate of 20 000 kg ha$^{-1}$.



**Experimental design and procedures**

In the present study, two tillage systems, conventional and conservation, were selected based on the widespread tillage practices in the study region (Jia et al., 2019; Wang, 2015; Zhang et al., 2015), and which also are applied globally (Liniger et al., 2017; Montgomery, 2017). The two conventional tillage practices included flat-planting without ridges and mulching (control, CK) and vertical ridging without mulching (Vr). The four conservation tillage measures included flat-planting and mulching without ridges (Cm, similar to no-till to some extent, Goddard et al., 2008), horizontal (contour) ridging without mulching (Hr), horizontal ridging with mulching (Hr+Cm), and vertical ridging with mulching (Vr+Cm). All plots were randomly arranged (Figure 2).

In terms of rainstorm status, generally momentary rainfall intensities larger than 23.4 mm h$^{-1}$ cause soil erosion with an approximate duration of 1 h in Northeast China (Zhang et al., 1992). In the present study, two rainfall intensities, 50- and 100-mm h$^{-1}$, lasting 1 h, were used as representative rainfall intensities (Xu et al., 2018; Wang et al., 2021a).

All plots were subjected to a pre-rain at 30 mm h$^{-1}$ for 5 min to ensure consistent soil moisture during experiments, consolidate loose soil particles, and flatten the soil surface, 24 h before experiments; rainfall intensity was calibrated to ensure the achievement of target intensity and fulfillment of experimental requirements (uniformity ≥90%, Figure 3a) before the experiment (Zhang et al., 2009b). After each rainfall event, the plots were restored via a process including drying, replacement and recovery of the topsoil layer and lost cornstalk, smashing of soil clods, restoring broken ridges, and smoothing of irregularities on the surface (Polyakov and Nearing, 2003).

**Experimental measurements**

**Runoff process**

Runoff-yielding time was measured using a stopwatch. Runoff velocity was measured thrice for each rainfall intensity in three soil sections (2, 5, and 7 m from the topsoil) after the runoff became steady, using the KMnO$_4$ dye tracer method (Zhang et al., 2009b).





**Runoff and soil loss**

Runoff and sediment samples were collected in 15-L buckets every 5 min once runoff occurred during each rainfall event. After allowing sediment settling for 1 h, the volume of supernatant was measured to calculate runoff loss. The sediment samples were oven-dried at 45 °C and weighed to calculate sediment yield and runoff rate.

**Soil splash-erosion**

Standard Morgan field splash cups (Morgan, 1978) were used to measure soil splash transport extent. Soil splash detachment was measured using specially designed aluminum cylindrical splash cups with 3-cm depth, 6-cm diameter, and a multihole bottom. The undisturbed soil was cut and packed into the cups and weighed immediately after drying at 45 °C. The soil cups were allowed to absorb moisture at 20-25 °C for 24 h. Three Morgan cups were arranged into each plot on the top-, mid-, and lower-slopes (at distances of 2, 5, and 7 m from the top), together with the small cups, as in Figure 3 (b, c). Rainfall was applied for 15 min to allow splash-erosion to occur. The soil was again weighed immediately after drying, and the splash transport and detachment amounts measured.

**Data analysis**

All data were analyzed for statistical significance of treatment effects by one-way analysis of variance (ANOVA) using SPSS 16.0 (SPSS Inc., Chicago, IL, USA). The least significant difference (LSD) at $p<0.05$ was used to compare the treatment means. Plots were drawn using Origin 9.0 (Origin Lab Corporation, Northampton, MA, USA).

**Results**

**Raindrop energy and distribution above/below corn seedling canopy**

As shown in Tables 1 and 2, the energy and size distribution of raindrops were significantly different between above and below the canopy of seedling corn. Under the two rainfall intensities, the canopy mitigation of raindrop energy was observed more in conservation than conventional tillage



measures. The percentage of raindrops with less than 2.5 mm diameter decreased when the raindrops
larger than 2.5 mm diameter decreased at the rainfall intensity of 50 mm h$^{-1}$, whereas the percentage
of raindrops with less than 2.0 mm diameter decreased when that of raindrops larger than 2.0 mm
diameter increased at the rainfall intensity of 100 mm/h.
**Runoff-yielding time and runoff velocity**

Table 3 shows that conservation tillage measures could significantly delay the runoff-yielding time

and decrease surface flow velocity, compared to CK and Vr, at the maize seedling stage. Compared
with CK and Vr, the runoff-yielding times of the Cm, Hr, Hr+Cm, and Vr+Cm treatments were
significantly postponed; the runoff-yielding time advanced at 100 mm h$^{-1}$ than at 50-mm h$^{-1}$. The
Hr+Cm treatment successfully prevented runoff yielding throughout the rainfall event under 50 mm h$^{-1}$
, and the average prolonged runoff-yielding time was approximately 26.1 min, which was 23.8-fold
that of the CK treatment under 100 mm h$^{-1}$. The average delay time durations for other treatments were
23.6 min for Hr, 5.6 min for Cm, and 2.8 min for Vr+Cm.

Table 3 also shows that the declining effects on surface flow velocity were more obvious under

light than under heavy rainfall intensity. Compared to the CK, the Hr+Cm, Cm, Vr+Cm, and Hr
treatments reduced the surface flow velocity significantly, with a decline of 100% (no runoff
generation), 75.8%, 71.9%, and 83.5%, respectively, at a rainfall intensity of 50 mm h$^{-1}$, and 96.4%,
82.9%, 77.7%, and 71.5%, respectively, at the rainfall intensity of 100 mm h$^{-1}$. However, Vr
significantly increased the runoff velocity by 50.3% and 10.1% at the rainfall intensities of 50 and 100
mm h$^{-1}$, respectively.
**Total runoff and soil loss**
**Surface runoff**

The conservation tillage measures of Cm, Hr, and Hr+Cm significantly reduced the runoff amount

compared to CK under the two rainfall intensities at the maize seedling stage (Figure 4). Compared to
CK, the Cm and Hr treatments reduced the runoff amount by 70.5% and 87.8%, respectively, at 50



mm h$^{-1}$ and by 44.8% and 58.9%, respectively, at 100 mm h$^{-1}$, respectively. The Hr+Cm treatment
entirely prevented runoff generation at 50 mm h$^{-1}$ and was still effective at 100 mm h$^{-1}$, restricting the
total runoff amount to a very low level of 20.79 L, accounting for only 16.6% of CK, and even causing
ridge rupture. The Vr+Cm treatment significantly decreased the runoff amount by 54.6% compared to
CK at 50 mm h$^{-1}$, but there was no difference at 100 mm h$^{-1}$. Conversely, Vr significantly enhanced
the runoff amount by 25.0% compared to CK at 50 mm h$^{-1}$, but there was no difference at 100 mm h$^{-}$
$^{1}$.

**Soil loss**

The total soil loss in Cm, Hr, Vr+Cm, and Hr+Cm was significantly lower than CK at the maize
seedling stage (Figure 5). Vr significantly augmented the soil loss amount by 7.03- and 2.29-fold at
the rainfall intensities of 50 and 100 mm h$^{-1}$, respectively. However, the total soil loss of CK was
greater than that of Cm, Hr, and Vr+Cm, exceeding by 11.9-, 6.0-, and 7.8-fold at 50 mm h$^{-1}$ and by
11.1-, 4.4-, 16.2-, and 20.5-fold at 100 mm h$^{-1}$, respectively. Like the effect on runoff amount, Hr+Cm
also showed the best performance for preventing runoff and soil loss at 50 mm h$^{-1}$ (Table 3 and Figure
4). The total soil loss was not different among the other three conservation measures of Cm, Hr, and
Vr+Cm at 50 mm h$^{-1}$, although the ridges of Hr were breached; meanwhile, Cm, Vr, and Hr+Cm
showed no significant difference, but Hr showed a significantly different soil loss from the three
treatments because of ridge rupturing at 100 mm h$^{-1}$. The results indicated that the conservation
measures were useful in reducing soil loss; in particular, mulching was more effective than contour
ridging, as seen in the case where the soil loss caused by Hr increased more than that caused by other
conservation measures, especially under high rainfall intensity conditions, when contour ridges were
destroyed.

**Horizontal ridge rupture**

As shown in Figures 6 and 7, mulching could not totally prevent contour ridge rupture, especially
under heavy rainfall conditions; for example, the ridge of Hr was destroyed at both rainfall intensities,



while that of Hr+Cm occurred only at 100 mm h$^{-1}$. The ridge rupture occurred earlier at 100 mm h$^{-1}$
than at 50 mm h$^{-1}$. The averaged runoff rate of Hr was 3.8-fold greater after ridge rupture than before
at 50 mm h$^{-1}$, being 22.6- and 1.6-fold greater under Hr and Hr+Cm at 100 mm h$^{-1}$, respectively.
Meanwhile, the average soil loss rate of Hr was 13.8-fold greater after ridge rupture than before at 50
mm h$^{-1}$, being 94.7- and 1.1-fold greater under Hr and Hr+Cm at 100 mm h$^{-1}$, respectively.
**Erosion process**
**Surface runoff process**
The runoff trends in most treatments were similar at both 50 and 100 mm h$^{-1}$ (Figure 6), including
two stages: 1) a low starting rate followed by a dramatic increase during the initial runoff-yielding
period, and 2) a relatively stable rate that persisted until the end of rainfall experiment. However, the
regular trends could be interfered with by a ridge rupture in the Hr and Hr+Cm treatments, with runoff
rates suddenly rising in the Hr-treated plot at 40 and 25 min under the rainfall intensities of 50 and 100
mm h$^{-1}$, respectively, and in the Hr+Cm treatment at 60 min under 100 mm h$^{-1}$ rainfall. In comparison,
the average runoff rate of CK was greater than that of Cm, Vr, Hr, and Vr+Cm by 2.9-, 0.8-, 5.0-, and
1.9-fold at 50 mm h$^{-1}$, respectively, and by 1.8-, 1.0-, 1.7-, and 1.2-fold, respectively, at 100 mm h$^{-1}$.
In addition, the average runoff rate of CK was 3.7-fold greater than that of Hr+Cm at 100 mm h$^{-1}$.
Compared to CK, the Cm, Hr, and Hr+Cm treatments reduced the runoff loss rates significantly
on all points within the entire rainfall experiment (Figure 6). At 50 mm h$^{-1}$, Hr showed a better capacity
for controlling runoff loss rates than Cm. Vr had no notable effects on runoff loss rates at most of the
points at 100 mm h$^{-1}$ but could promote the loss rate significantly at 50 mm h$^{-1}$, including the whole
process except for the runoff-yielding point. The runoff loss rates of Vr+Cm were significantly lower
than those of CK at 50 mm h$^{-1}$, with an average runoff rate of 53.6%, while the reduction was very
limited at 100 mm h$^{-1}$.
Figure 6 also illustrates that the stable runoff rates were lower at 50 mm h$^{-1}$ than at 100 mm h$^{-1}$ in
all treatments. The runoff rates of CK, Cm, Vr, Hr, Vr+Cm, and Hr+Cm stabilized at approximately



91.8, 30.1, 118.7, 20.3, 48.2, and 0 mL s$^{-1}$ at 50 mm h$^{-1}$, respectively, and at 198.6, 117.4, 192.5, 122.9,
176.1, and 49.9 mL s$^{-1}$ at 100 mm h$^{-1}$, respectively.

The results suggested that the mulching treatments, including Cm, Hr+Cm, and Vr+Cm, could

mitigate rate-changing magnitudes compared to the corresponding tillage measures without mulching,
that is CK, Hr, and Vr, indicating that more rainfall was infiltrated or stored under the treatments with
mulching compared to those without mulching.

**Sediment yielding process**

As shown in Figure 7, the sediment loss rates in most treatments varied based on the changing

trends of the runoff loss rate (Figure 6), with a relatively low starting level and then varied within a
certain range based on rainfall intensity. The four conservation practices could effectively reduce soil
loss rate compared to the conventional tillage of CK and Vr, except that the ridges ruptured, and the
Vr treatment obviously enhanced the soil loss rate compared to CK. In comparison, the average soil
loss rates of CK were 10.0-, 3.7-, and 6.6-fold greater than those of Cm, Hr, and Vr+Cm at 50 mm h$^{-1}$,
respectively, and 13.0-, 3.0-, 16.2-, and 12.6-fold greater than those of Cm, Hr, Vr+Cm, and Hr+Cm
at 100 mm h$^{-1}$, respectively. However, the averaged soil loss rates of Vr were 7.0- and 2.3-fold greater
than those of CK at 50- and 100-mm h$^{-1}$, respectively.

The impact of ridge rupture was greater at 100 mm h$^{-1}$ than at 50 mm h$^{-1}$, and the subsequent soil

loss rates would stay higher thereafter, rather than being at the former level at 100 mm h$^{-1}$, which
dropped to former rates under 50 mm h$^{-1}$ (Figure 5). Hr could reduce the sediment loss rate throughout
the entire rainfall process, averaging 82.0% and 68.40% of CK under the two rainfall intensities, but
two of the three ridge rupture time points made the instantaneous rates higher than the earlier rates.

During rainfall events, the mean soil loss rates in the three mulching treatments of Cm, Vr+Cm,

and Hr+Cm were approximately 0.01, 0.02, and 0 g s$^{-1}$ at 50 mm h$^{-1}$, and 0.09, 0.07, and 0.09 g s$^{-1}$ at
100 mm h$^{-1}$, respectively, being significantly lower than those of CK, which were approximately 0.15
and 1.18 g s$^{-1}$ at 50 and 100 mm h$^{-1}$, respectively. The soil loss rates of these mulching treatments were





also lower than those of the non-mulching treatments, such as Vr and Hr, which were approximately
1.02 and 0.04 g s$^{-1}$ at 50 mm h$^{-1}$ and 2.70 and 0.39 g s$^{-1}$ at 100 mm h$^{-1}$, respectively (Figure 7). Mulching
also mitigated the changing trends of sediment loss rate, i.e., restricting the rate variation magnitude
to a lower scale. Therefore, the mulching treatments were more effective in controlling the sediment
yield compared to no mulch treatments.

**Factors influencing soil loss**

The relationship between sediment yield and splash-detachment, splash-transport, total runoff, and
surface flow rate was analyzed, and are illustrated in Figure 8 and Table 4. The mulching treatments
could restrict splash-erosion to very low levels, reducing the average splash-detachment and splash-
transport amounts from 143.16 to 1.13 g m$^{-2}$ h$^{-1}$ and from 1063.90 to 8.93 g m$^{-2}$ h$^{-1}$, respectively. The
ridge treatments had no significant impacts on splash-erosion. Thus, for uncovered plots, splash
erosion was mainly influenced by rainfall intensity. The linear correlation coefficients ($R^2$) of the
splash-detachment and splash-transport rates to rainfall intensity were 0.93 and 0.98, respectively. The
splash rates of Cm were also partly related to the rainfall intensity, but the correlation was more
complicated, and thus further study is needed.
In general, the total soil loss increased with an increase in splash-erosion rate, escalating in non-
mulching treatments under light rainfall conditions. However, when the plots suffered ridge rupture,
the impact of splash-erosion on soil loss appeared to be insignificant. With an increase in runoff
volume and velocity, soil loss would also ascend, and thus treatments with high runoff volume and
velocity would also lead to serious soil loss. However, this regulation was not applicable to mulching
treatments.

**Discussion**

**Effects of tillage measures on runoff**

We verified that crops could act as a type of vegetation cover (Table 1 and 2) and play an important
role in mitigating runoff and soil loss on sloping farmlands, in agreement with previous studies (Cerdà



et al., 2017; Prosdocimi et al., 2016a, b; Wang et al., 2018). Different tillage systems have different
impacts on soil erosion associated with processes occurring in slope farmlands (Liu et al., 2011; Xu et
al., 2018). The Vr treatment has already been verified to increase soil erosion because of
microtopography changes (Liu et al., 2011; Zhang et al., 2009a).

In the present study, conservation tillage could significantly postpone runoff yield and decrease

runoff velocity compared to conventional tillage. Our results indicated that horizontal ridges,
mulching, or seedling corn canopy were effective in controlling runoff generation, especially at 50
mm/h, at the maize seedling stage. The conservation measures could have enhanced the infiltration
capacity of water or increased soil surface roughness (Rodríguez-Caballero et al., 2012; Vermang et
al., 2015; Wang et al., 2018), and crop leaves could intercept rainfall and alter raindrop diameter and
energy (Ma et al., 2013; Zhang et al., 2015). As there are only limited chances for extreme precipitation
in the region (Zhang et al., 2010), adopting Hr and Cm would limit runoff generation. In addition, the
two tillage measures also reduced the runoff-flow velocity, which is a key factor influencing runoff
energy and erosiveness (Vermang et al., 2015); both Hr and Cm performed better at 50 mm h$^{-1}$ than at
100 mm h$^{-1}$. Our results are consistent with previous studies on other soil types (Prosdocimi et al.,
2016a, b; Xu et al., 2017). The runoff generation was postponed and the surface-flow velocity
decreased mainly because both Hr and Cm treatments changed the microtopography of the soil with
increasing surface roughness (Vermang et al., 2015; Wang et al., 2018) and the infiltration of
conservation tillage was higher than that of conventional measures. The outcome offered more water
storage microstructure for the surficial soil, causing the rainwater to infiltrate rather than flowing
downhill (Liu et al., 2015; USDA-ARS, 2008, 2013). The outcome also increased the friction between
rainwater and land, thereby reducing runoff velocity. Comparing the effects of Hr and Cm, Hr set a
higher threshold for runoff yield, as it could lead to more water storage between ridges. However, once
the runoff had occurred, Cm performed better, since the presence of cornstalk could reduce the flow
velocity to a very low level. Thus, Hr+Cm is the optimal treatment from the perspective of postponing



runoff-yield and restricting the destruction of runoff, once generated.
The runoff loss rate significantly increased following a low start during the runoff generation
period and then remained stable at a certain level, based on the rainfall intensity. The results correspond
with the findings of a study in purple soil (Xu et al., 2008). Hr and Cm could effectively constrain the
runoff loss rates and decrease the runoff amount, especially at 50 mm h$^{-1}$. The Hr+Cm treatment, which
combined horizontal ridging and mulching, influenced runoff under all rainfall types, especially under
a rainfall intensity of 50 mm h$^{-1}$. As runoff is the main vector affecting both soil loss and agricultural
non-point source pollution (Hudson, 2015; Zhang et al., 2007), Hr+Cm should be recommended as an
effective tillage practice in the region.
However, this recommendation would engender extremely higher outliers for runoff rate as a real-
time response to ridge rupture when the plots were treated with Hr, especially under heavy rainfall
conditions (Li et al., 2016; Lu et al., 2016). In this case, the water held by the two adjacent ridges
drained immediately after ridge rupture and rushed out into the next inter-ridge area, causing either
successive ridge ruptures or runoff overflow, both of which could prompt a sudden upsurge in runoff
rate (Xu et al., 2018). Consequently, the total runoff loss amount also increased. The rising magnitude
caused by ridge rupture depended on the rupture time and location of the initially ruptured ridge. In
the present study, in the Hr-treated plot, ridge rupture occurred relatively earlier and closer to the top
of the plot under a rainfall intensity of 100 mm h$^{-1}$ than under 50 mm h$^{-1}$ resulting in greater runoff
loss. Thus, enhancing the quality of ridges to improve their water pressure tolerance capability is vital
when applying horizontal ridges (Liu et al., 2014a).
Mulching could directly lead to water absorption and protection of a ridge from saturation and
erosion by raindrops and runoff (Cerdà et al., 2016; Jordán et al., 2010), thereby reducing the risk of
ridge rupture. In the present study, Hr-treated plots suffered three times as many ridge ruptures, while
the Hr+Cm plots suffered only one ridge rupture. Moreover, no successive ridge ruptures were
observed in the Hr+Cm plots, because mulching and soil blocks would likely be obstructed by the next





ridge with the presence of cornstalk, rather than triggering successive ridge ruptures, even if one of the
ridges happened to rupture. Moreover, ridge-furrow planting under mulching conditions played an
effective role in reducing surface runoff with an increase in soil-water infiltration (Gholami et al.,
2013; Kader et al., 2017).
Vr could increase the runoff loss rate and amount under light rainfall conditions, as shown by Shen
et al. (2005) and Zhang et al. (2009a) on black soil, and by Xu et al. (2008) on purple soil farmlands
compared to the runoff between contours and downslope ridges. Therefore, vertical ridges should be
avoided on slope croplands in the region.
**Effects of tillage measures on soil loss**
Both Hr and Cm could alleviate soil erosion, mainly by improving the microtopography to increase
soil surface roughness (Rodríguez-Caballero et al., 2012; Vermang et al., 2015), and improve soil
physicochemical properties. Moreover, Vr should be circumvented as it augments both soil loss rate
and amount (Kader et al., 2017; Mulumba and Lal, 2008).
When there was no ridge rupture during the rainfall, Hr effectively reduced sediment yield and
soil loss rate, as shown in previous studies (García-Orenes et al., 2012). However, after ridge rupture,
the impacts on sediment loss were much more severe than on runoff, e.g., the runoff rate was amplified
22.6 times compared to its neighboring point, while the sediment loss rate was amplified 94.7 times
after ridge rupture occurred in Hr under a rainfall intensity of 100 mm h$^{-1}$. This outcome may have
occurred because the broken ridges, which were normally big soil blocks, were prone to being directly
swept and, thus, lost via runoff (Xu et al., 2018). The residual ridge remaining to be washed
continuously by runoff would also increase the sediment concentration in runoff after the ridge rupture,
leading to a higher soil loss rate. Soil loss would be further amplified if ridge rupture occurred in the
top section of the plot and thus likely triggered successive ruptures.
Our study revealed that Cm was more reliable than Hr in controlling soil loss (Kader et al., 2017;
Prosdocimi et al., 2016b), as it could restrict both the sediment yield and soil loss rate to very low





levels (García-Orenes et al., 2012). The reason might be that the flow could accumulate sufficient
power to detach and transport particles with mulching (Mannering and Meyer, 1963; Poesen and
Lavee, 1991). In addition, Cm could postpone the soil loss rate that increasingly responded to rainfall
intensity enhancement, which is an important effect on soil erosion because rainfall has a short duration
but high intensity during the maize seeding stage in Northeastern China (Sun et al., 2000; Zhang et al.,
2010). This postponing effect would counteract or even eliminate the instantaneous serious destruction
due to torrential rain. Hence, Hr+Cm significantly prevented soil loss, especially under light rainfall
intensity conditions, and thus, in practice, should be suggested to reduce soil erosion.
**Influencing factors**
Soil erosion is related to both runoff strength and soil erodibility (Tang, 2004; Wang et al., 2012;
Wang, 1993). Runoff serves as a vector for sediment (Hudson, 2015), and the final sediment yield is
based on both runoff strength and soil erodibility (Wang, 1993). Runoff strength can be illustrated by
volume and velocity, representing its amount and energy, respectively (Prosdocimi et al., 2016a).
Generally, in our study, the treatments with higher runoff strength experienced worse soil erosion.
However, grievous splash-erosion, i.e., worse erodibility, did not always correspond to high soil loss.
Therefore, runoff strength should be a direct predictor of soil erosion.
According to our results, higher-strength runoff and more soil loss was observed with heavier
rainfall, which indicated that the hydrological response of the soil is based on Hortonian flow type
(Bombino et al., 2021).
At the seedling stage, maize plants could protect the surface soil from splash-erosion by preventing
direct raindrop action, reducing their kinetic energy, and by changing the distribution of raindrops
because of canopy gaps (Ghahramani et al., 2011; Miyata et al., 2009). Nevertheless, as discussed
earlier, splash-erosion has a limited influence on total soil loss amount. Therefore, the excellent effects
of mulching on erosion control shown in this experiment should mainly result in two other functions,
reducing runoff strength and filtering out runoff soil particles (Prosdocimi et al., 2016a, b). Both





functions caused a reduction in sediment concentration because of the effects of mulching as buffer
strips (Fang, 2017).
**Horizontal ridge rupture**

Horizontal ridge rupture or breaching is a common concern in Northeast China, as erosive storms

can occur in summer with short duration but high intensity (Shen et al., 2005); such storms often
coincide with snowmelt runoff in spring (Li et al., 2016; Lu et al., 2016; Xu et al., 2018). Contour
ridge stability is mainly related to ridge geometry, sloping land microtopography, soil physical
properties of the ridge body, and rainfall characteristics (Liu et al., 2014a; Shen et al., 2005). In
addition, the sediment concentration stayed higher theafter rather than being at the former level at 100
mm h$^{-1}$, while dropping to former rates under 50 mm h$^{-1}$(Fig. 7), which might be due to the significant
differences in runoff, sediment, and infiltration amount under the two rainfall intensities (Liu et al.,
2014a; Liu et al., 2019; Shen et al., 2005).

Generally, Hr can increase water infiltration before breaching (Liu et al., 2015; USDA-ARS, 2008,

2013) and lead to abundant sediment storage (Xu et al., 2018). Time of ridge rupture shortens with
higher rainfall intensity (Liu et al., 2015; Liu et al., 2014a; Liu et al., 2014b; Xu et al., 2018). Extremely
high runoff and soil loss rates after rupture are analogous to the relationships among the peaks of runoff
and sediment yield and ridge failure (Liu et al., 2015; Liu et al., 2014b; Xu et al., 2018). Averaged
peak runoff and soil loss rates after ridge failure were 9.3- and 36.7-fold those prior neighboring points,
respectively. The ratio of peak sediment rate to base sediment rate under Hr in this study ranged from
13.8 to 94.7 g L$^{-1}$. The varied range differed but included previous results reported by Liu et al. (2014b)
and Xu et al. (2018). Our study showed that contour ridges rupturing at 50 mm h$^{-1}$ were not in
agreement with the results of Xu et al. (2018), possibly because of the differences in ridge geometry
characteristics, such as ridge height. Liu et al. (2014b) suggested that increasing ridge height might
prevent horizontal ridge failure and decrease soil loss hazard risk, considering enhanced water storage
capacity.



Our study illustrated that mulching could not always avert ridge rupture but could significantly
postpone the collapse time of ridge failure (Figure 6 and 7), possibly because mulching improves soil
properties (Kader et al., 2017; Kurothe et al., 2014; Prosdocimi et al., 2016a, b) and, therefore, alters
runoff and soil erosion characteristics (Gholami et al., 2013).
**Conclusions**
Rainfall simulation experiments were conducted to study the effects of six measures of two tillage
systems on water-based soil erosion of a black soil hillslope during the maize seedling stage under two
rainfall intensities (50 and 100 mm h$^{-1}$) in Northeast China. The results showed that corn seedlings
could protect the surface soil from splash-erosion by reducing the kinetic energy and changing the
distribution of raindrops. Conservation measures with mulching significantly reduced water and soil
loss compared to conventional tillage. Mulching had an ideal erosion-controlling capacity. In addition,
mulching could mitigate soil loss increase caused by heavy rainfall. The positive effects of mulching
were based on its strong ability to reduce splash-erosion and runoff volume and, more importantly, on
its function to decrease runoff velocity and filter runoff sediment in. Vr further exacerbates soil erosion
and should normally be avoided. The horizontal ridging plus mulching treatment had the optimal
performance and should be adopted as an optimized tillage measure in black soil hillslope to restrict
soil erosion in corn seedling stage.

**Funding**
This work was supported by the National Natural Science Foundation of China (No. 32272819), and
the Jilin Scientific and Technological Development Program (No. 202203022003NC and No.
20210202011NC). Special thanks are owed to the anonymous reviewers and editors.
**Code/Data availability**
The original contributions presented in the study are included in the article/Supplementary Material;
further inquiries can be directed to the corresponding author.





**Author contribution**
NC and YBZ designed the research and supervised the project. YCW, ZL, LSW, BL, and LYH were
key players for the field trials and collected data. YCW, ZL, and YZ analyzed the data and verified the
analytical methods. DYG, YBZ, NC, and JHC wrote the manuscript.
**Competing interests**
The authors declare that the research was conducted in the absence of any commercial or financial
relationships that could be construed as a potential conflict of interest.

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





**Figure Legends**
**Figure 1.** Field scenario at the maize seedling stage in the Mollisols of Northeast China.

**Figure 2**. Experimental plots, status, and rainfall setup.

**Figure 3**. Rainfall intensity calibration and small splash-cup positions. (a) Rainfall intensity calibration
performed every time before rainfall experiment. (b) Positions for small splash-cups in plots with
vertical ridges. (c) Positions for small splash-cups in plots with horizontal ridges.

**Figure 4**. Runoff amount under different tillage measures. Control (CK), flat-planting without ridges
and mulching; Hr, horizontal ridging without mulching; Vr, vertical ridging without mulching; Cm,
flat-planting and mulching without ridges; Hr+Cm, horizontal ridging with mulching; Vr+Cm, vertical
ridging with mulching. The vertical error bars indicate LSD at $P<0.05$. Note: The asterisk (*) indicates
ridge rupture.

**Figure 5**. Soil loss amount under different tillage measures. Control (CK), flat-planting without ridges
and mulching; Hr, horizontal ridging without mulching; Vr, vertical ridging without mulching; Cm,
cornstalk mulching; Hr+Cm, horizontal ridging with mulching; Vr+Cm, vertical ridging with
mulching. The vertical error bars indicate LSD at $P<0.05$. Note: The asterisk (*) indicates ridge
rupture.

**Figure 6**. Runoff rate under different tillage measures. Control (CK), flat-planting without ridges and
mulching; Hr, horizontal ridging without mulching; Vr, vertical ridging without mulching; Cm,
cornstalk mulching; Hr+Cm, horizontal ridging with mulching; Vr+Cm, vertical ridging with
mulching.




**Figure 7**. Soil loss rate under different tillage measures. Control (CK), flat-planting without ridges and
mulching; Hr, horizontal ridging without mulching; Vr, vertical ridging without mulching; Cm,
cornstalk mulching; Hr+Cm, horizontal ridging with mulching; Vr+Cm, vertical ridging with
mulching.

**Figure 8**. Correlation between soil loss and influencing factors (a), correlation of soil loss amount and
soil splash-detachment; (b), correlation of soil loss amount and splash-transport amount; (c),
correlation of soil loss amount and runoff loss amount; d. correlation of soil loss amount and runoff
velocity. Note: Correlations between total soil loss amount and four inferred influencing factors; The
symbol ▲ indicates ridge rupture during the rainfall experiment.



Table 1. Effect of canopy on kinetic energy

| | 50 mm h⁻¹ | | 100 mm h⁻¹ | |
|---|---|---|---|---|
| | CM | CT | CM | CT |
| | kinetic energy, J/(m²·mm) | | | |
| above | 16.43 c | | 18.19 a | |
| below | 15.78 d | 15.84 d | 17.25 b | 17.38 b |
| | total kinetic energy, J/m² | | | |
| above | 196.5 d | | 407.64 a | |
| below | 174.05 e | 178.2 e | 357.97 c | 367.1 b |

CM, conservation tillage measures, including Cm, cornstalk mulching without ridges; Hr, horizontal
ridging without mulching; Vr+Cm, vertical ridging with mulching; Hr+Cm, horizontal ridging with
mulching. CT, conventional tillage practices, including control (CK), flat-planting without ridges and
mulching, and Vr, vertical ridging without mulching.
Values followed by different letters are significantly different at *P*<0.05 according to the LSD test.



Table 2
Effect of canopy on raindrop diameter

| Raindrop diameter, mm | 50 mm h$^{-1}$, % | | 100 mm h$^{-1}$, % | |
|---|---|---|---|---|
| | above | below | above | Below |
| 0.5–1 | 3.16 | 2.08 | 5.02 | 3.01 |
| 1–1.5 | 32.81 | 29.87 | 35.97 | 34.99 |
| 1.5–2.0 | 19.96 | 17.96 | 22.99 | 21.00 |
| 2.0–2.5 | 20.95 | 19.95 | 17.00 | 21.99 |
| 2.5–3 | 12.06 | 13.99 | 10.01 | 13.00 |
| 3–3.5 | 11.07 | 13.00 | 9.01 | 5.01 |
| 3.5–4 | 0 | 2.08 | 0 | 1.01 |
| 4–4.5 | 0 | 1.08 | 0 | 0 |




Table 3
Runoff-yielding time and runoff velocity under different tillage practices.

| Treatment | Runoff-yielding time (s) | | Runoff velocity ($10^{-2}$ m s$^{-1}$) | |
|---|---|---|---|---|
| | 50 mm h$^{-1}$ | 100 mm h$^{-1}$ | 50 mm h$^{-1}$ | 100 mm h$^{-1}$ |
| CK | 129 d | 69 e | 5.83 b | 17.95 a |
| Cm | 611 b | 260 c | 1.41 c | 3.07 c |
| Vr | 132 d | 71 e | 8.76 a | 19.77 a |
| Hr | 1700 a | 1332 b | 0.96 d | 5.12 b |
| Vr+Cm | 374 c | 154 d | 1.64 c | 4.01 b |
| Hr+Cm | NA | 1634 a | NA | 0.65 d |

CK, control, flat-planting without ridges and mulching; Cm, cornstalk mulching without ridges; Vr,
ridging without mulching; Hr, horizontal ridging without mulching; Vr+Cm, vertical ridging with
mulching; Hr+Cm, horizontal ridging with mulching; NA, Hr+Cm-treated plots prevented runoff
throughout the rainfall experiment.
Values in the same column followed by different letters are significantly different at $P<0.05$ according
to the LSD test.



Table 4
Splash-detachment and splash-transport under different tillage practices.

| Treatment | | 50 mm h⁻¹ | | | 100 mm h⁻¹ | | |
|---|---|---|---|---|---|---|---|
| | | Splash-detachment, g/m² | Splash-transport, g/m² | Ratio of transport, % | Splash-detachment, g/m² | Splash-transport, g/m² | Ratio of transport, % |
| Conventional tillage | CK | 377.55 | 40.39 | 10.70 | 1750.25 | 245.94 | 14.05 |
| | Vr | 386.13 | 36.69 | 9.50 | 1695.67 | 212.93 | 12.56 |
| Conservation tillage | Cm | 7.97 | 0.67 | 8.35 | 9.90 | 1.60 | 16.11 |
| | Hr | 369.24 | 43.18 | 11.69 | 1723.74 | 226.26 | 13.13 |
| | Vr+Cm | 6.16 | 0.76 | 12.31 | 11.63 | 1.97 | 16.93 |
| | Hr+Cm | 7.92 | 0.81 | 10.23 | 13.65 | 1.86 | 13.63 |

CK, control, flat-planting without ridges and mulching; Cm, cornstalk mulching without ridges; Vr,
ridging without mulching; Hr, horizontal ridging without mulching; Vr+Cm, vertical ridging with
mulching; Hr+Cm, horizontal ridging with mulching.



Table 5
Change in soil water content on soil profile pre- and post-rainfall and infiltration under different
tillage practices

| Treatments | | Depth, cm | 50 mm h$^{-1}$ | | | | 100 mm h$^{-1}$ | | | |
|---|---|---|---|---|---|---|---|---|---|---|
| | | | Soil water content, % | | | Infiltration, mm | Soil water content, % | | | Infiltration, mm |
| | | | Pre-rainfall | Post-rainfall | Rising rate, % | | Pre-rainfall | Post-rainfall | Rising rate, % | |
| Conventional tillage | CK | 0–5 | 21.22 | 25.04 | 17.99 | 26.4 | 25.17 | 30.19 | 19.90 | 36.69 |
| | | 5–10 | 26.59 | 28.19 | 5.99 | | 27.48 | 28.51 | 3.78 | |
| | | 10–20 | 22.15 | 22.33 | 0.81 | | 25.64 | 25.93 | 1.15 | |
| | Vr | 0–5 | 24.25 | 27.69 | 14.18 | 24.42 | 25.50 | 29.71 | 16.52 | 35.34 |
| | | 5–10 | 24.10 | 25.63 | 6.37 | | 29.54 | 33.24 | 12.53 | |
| | | 10–20 | 22.88 | 23.18 | 1.32 | | 27.67 | 28.31 | 2.32 | |
| Conservation tillage | Cm | 0–5 | 27.19 | 29.31 | 7.80 | 31.98 | 27.79 | 33.19 | 19.44 | 45.81 |
| | | 5–10 | 31.00 | 33.33 | 7.50 | | 27.89 | 30.29 | 8.59 | |
| | | 10–20 | 27.19 | 29.07 | 6.90 | | 25.55 | 27.04 | 5.81 | |
| | Hr | 0–5 | 27.56 | 35.67 | 29.42 | 44.16 | 23.64 | 32.69 | 38.30 | 65.58 |
| | | 5–10 | 27.62 | 32.12 | 16.30 | | 28.17 | 30.62 | 8.69 | |
| | | 10–20 | 25.22 | 27.65 | 9.64 | | 24.52 | 27.48 | 12.07 | |
| | Vr+ Cm | 0–5 | 28.54 | 32.65 | 14.39 | 33.18 | 29.20 | 34.74 | 18.96 | 44.28 |
| | | 5–10 | 31.39 | 34.69 | 10.51 | | 29.22 | 33.12 | 13.33 | |
| | | 10–20 | 23.45 | 25.94 | 10.62 | | 29.78 | 32.68 | 9.74 | |
| | Hr+ Cm | 0–5 | 27.70 | 35.28 | 27.38 | 44.76 | 28.13 | 36.54 | 29.90 | 71.64 |
| | | 5–10 | 30.11 | 34.18 | 13.52 | | 30.98 | 34.65 | 11.85 | |
| | | 10–20 | 25.34 | 29.81 | 17.64 | | 27.96 | 30.49 | 9.02 | |

CK, control, planted flat without ridges and mulching; Cm, cornstalk mulching; Vr, vertical ridges
without mulching; Hr, horizontal ridges without mulching; Vr+Cm, vertical ridges with mulching;
Hr+Cm, horizontal ridges with mulching.





**Figure 1.** Field scenario at the maize seedling stage in the Mollisols of Northeast China.

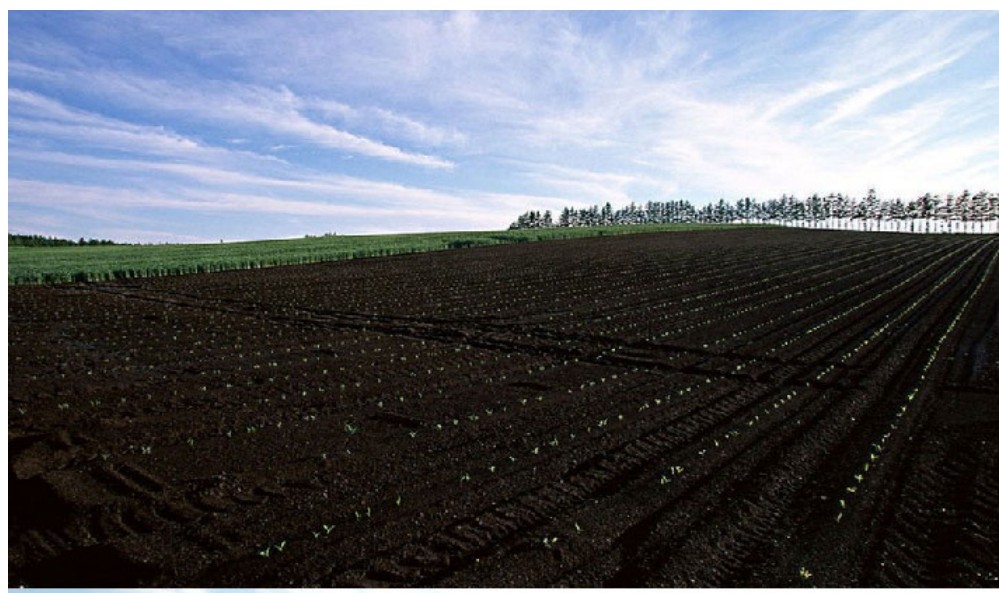

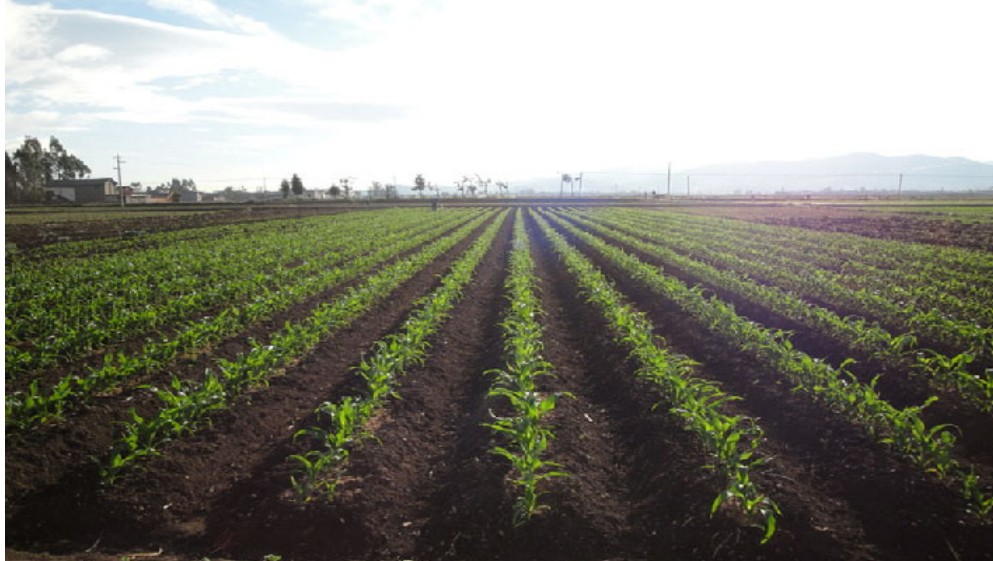








**Figure 2**. Experimental plots, status, and rainfall setup.

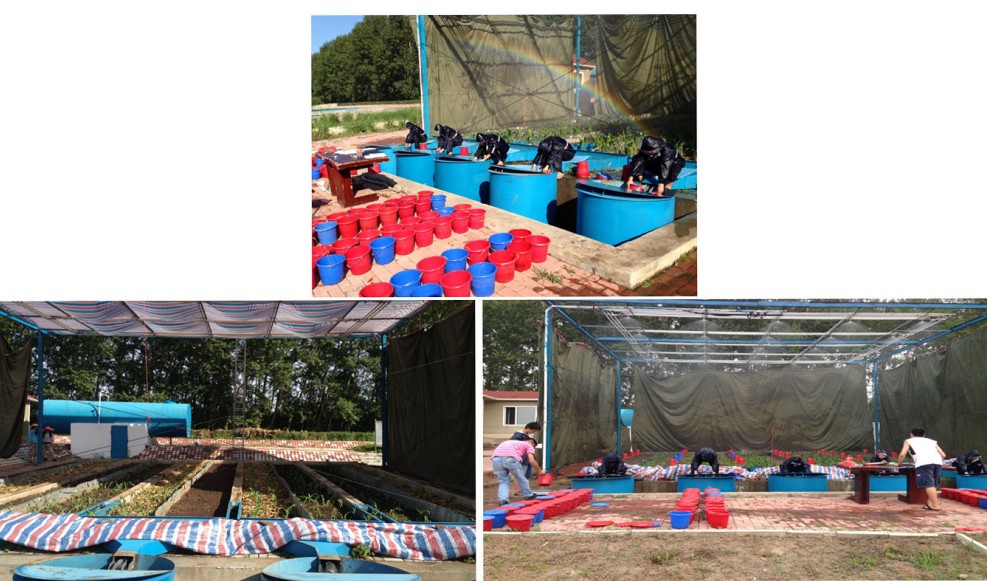


















**Figure 3**. Rainfall intensity calibration and small splash-cup positions. (a) Rainfall intensity
calibration performed every time before rainfall experiment. (b) Positions for small splash-cups in
plots with vertical ridges. (c) Positions for small splash-cups in plots with horizontal ridges.


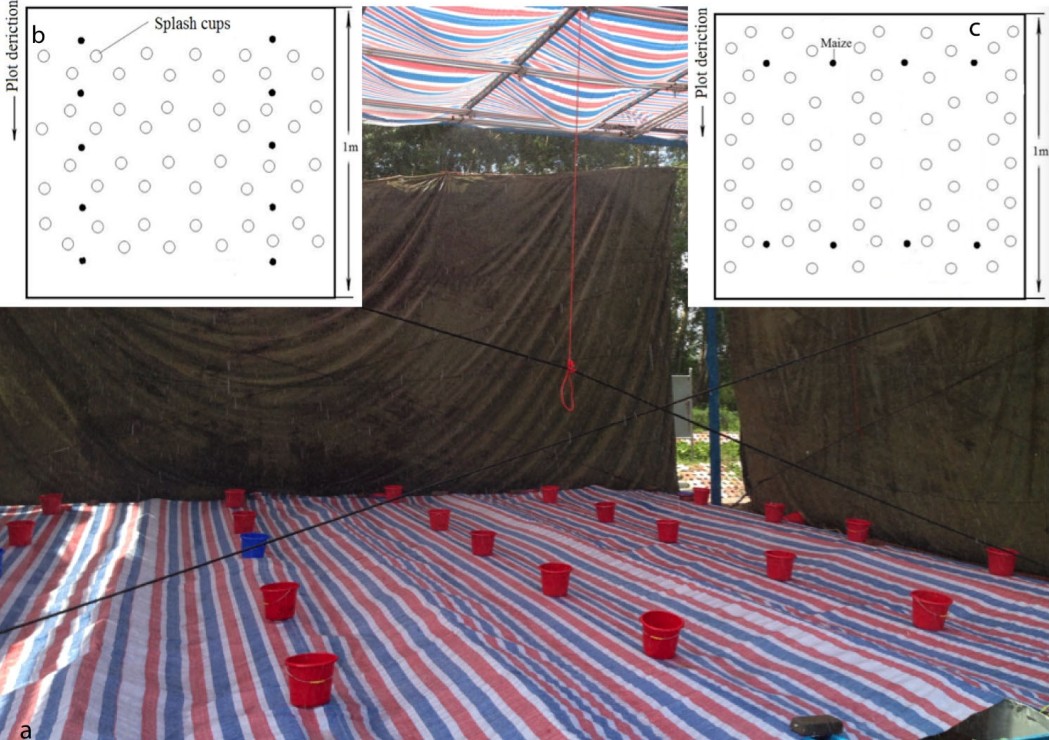











**Figure 4**. Runoff amount under different tillage measures. Control (CK), flat-planting without ridges
and mulching; Hr, horizontal ridging without mulching; Vr, vertical ridging without mulching; Cm,
flat-planting and mulching without ridges; Hr+Cm, horizontal ridging with mulching; Vr+Cm,
vertical ridging with mulching. The vertical error bars indicate LSD at $P<0.05$. Note: The asterisk (*)
indicates ridge rupture.

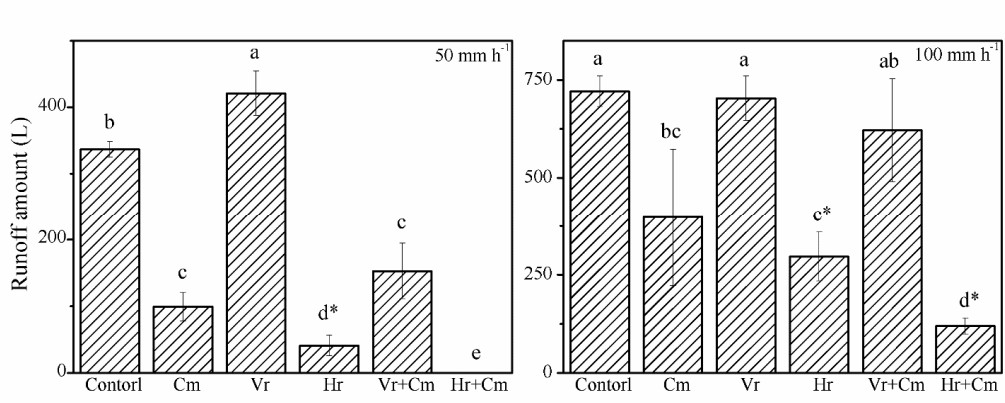










**Figure 5**. Soil loss amount under different tillage measures. Control (CK), flat-planting without
ridges and mulching; Hr, horizontal ridging without mulching; Vr, vertical ridging without mulching;
Cm, cornstalk mulching; Hr+Cm, horizontal ridging with mulching; Vr+Cm, vertical ridging with
mulching. The vertical error bars indicate LSD at $P<0.05$. Note: The asterisk (*) indicates ridge
rupture.

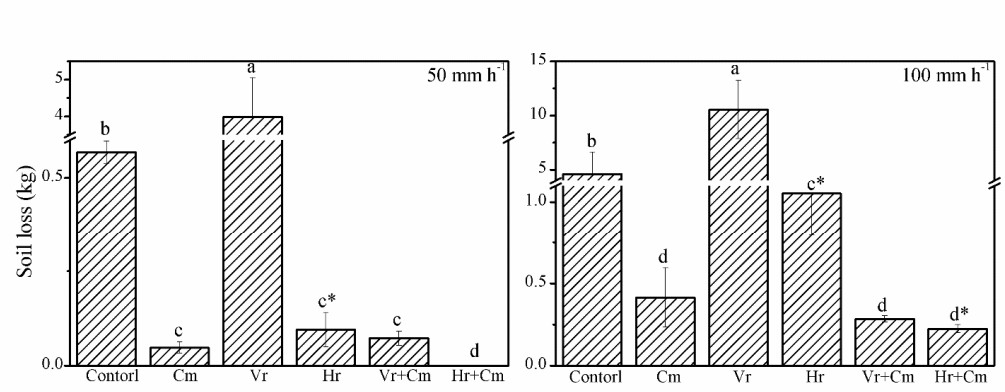











**Figure 6**. Runoff rate under different tillage measures. Control (CK), flat-planting without ridges and
mulching; Hr, horizontal ridging without mulching; Vr, vertical ridging without mulching; Cm,
cornstalk mulching; Hr+Cm, horizontal ridging with mulching; Vr+Cm, vertical ridging with
mulching.

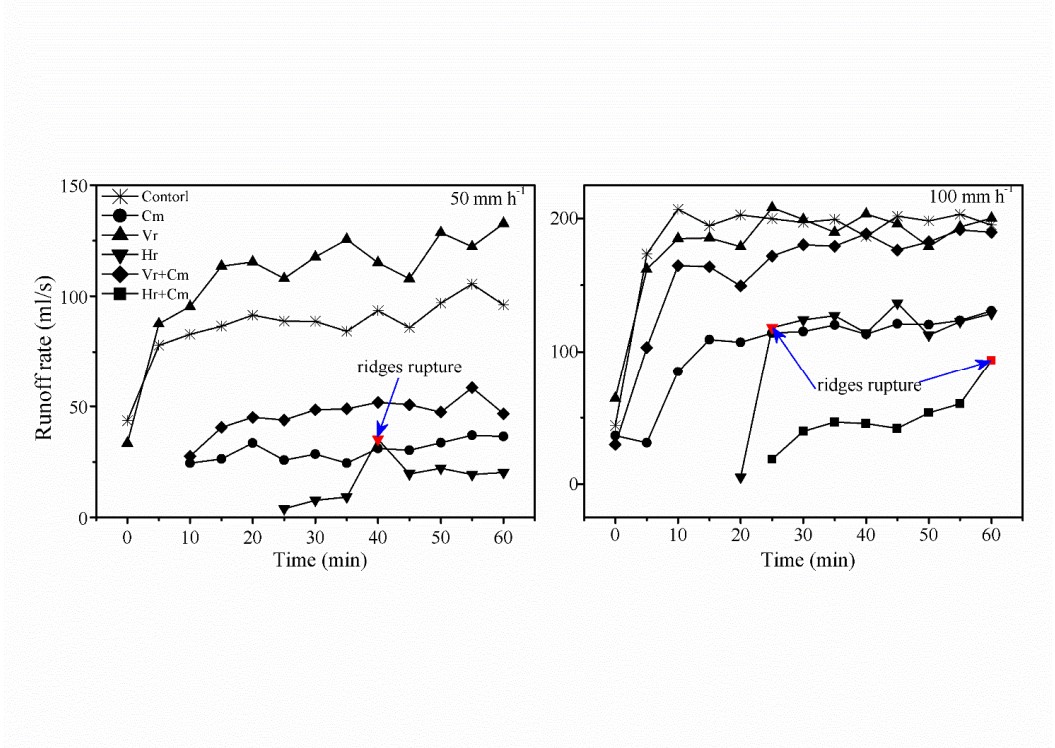













**Figure 7**. Soil loss rate under different tillage measures. Control (CK), flat-planting without ridges
and mulching; Hr, horizontal ridging without mulching; Vr, vertical ridging without mulching; Cm,
cornstalk mulching; Hr+Cm, horizontal ridging with mulching; Vr+Cm, vertical ridging with
mulching.

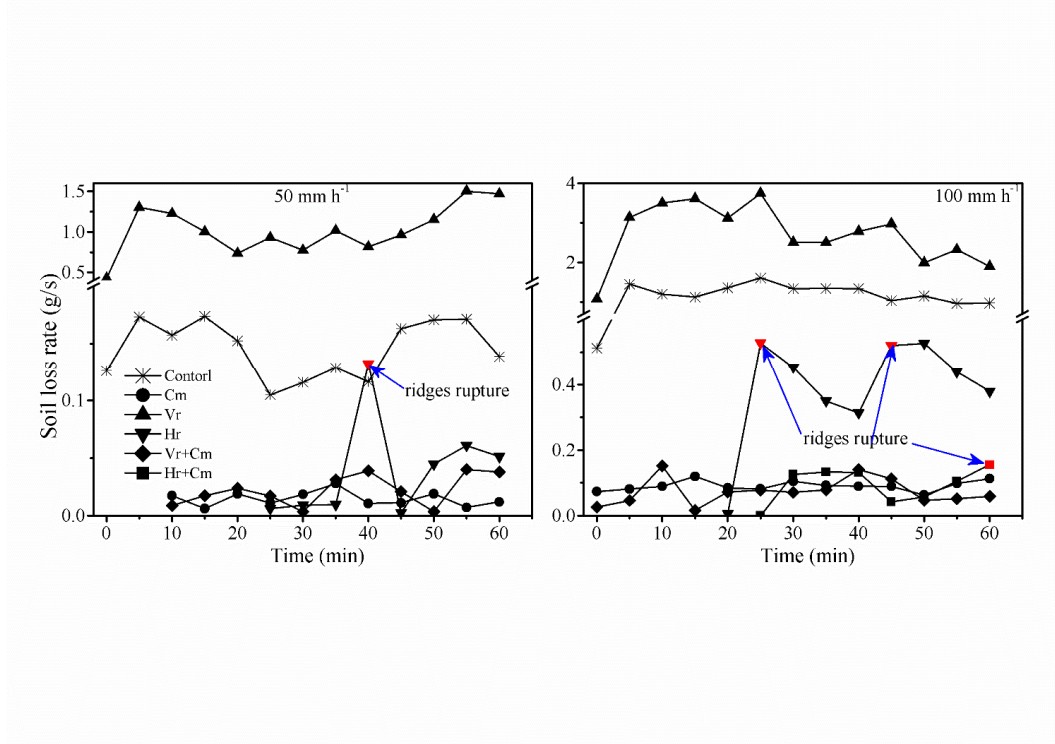











**Figure 8**. Correlation between soil loss and influencing factors (a), correlation of soil loss amount
and soil splash-detachment; (b), correlation of soil loss amount and splash-transport amount; (c),
correlation of soil loss amount and runoff loss amount; d. correlation of soil loss amount and runoff
velocity. Note: Correlations between total soil loss amount and four inferred influencing factors; The
symbol ▲ indicates ridge rupture during the rainfall experiment.

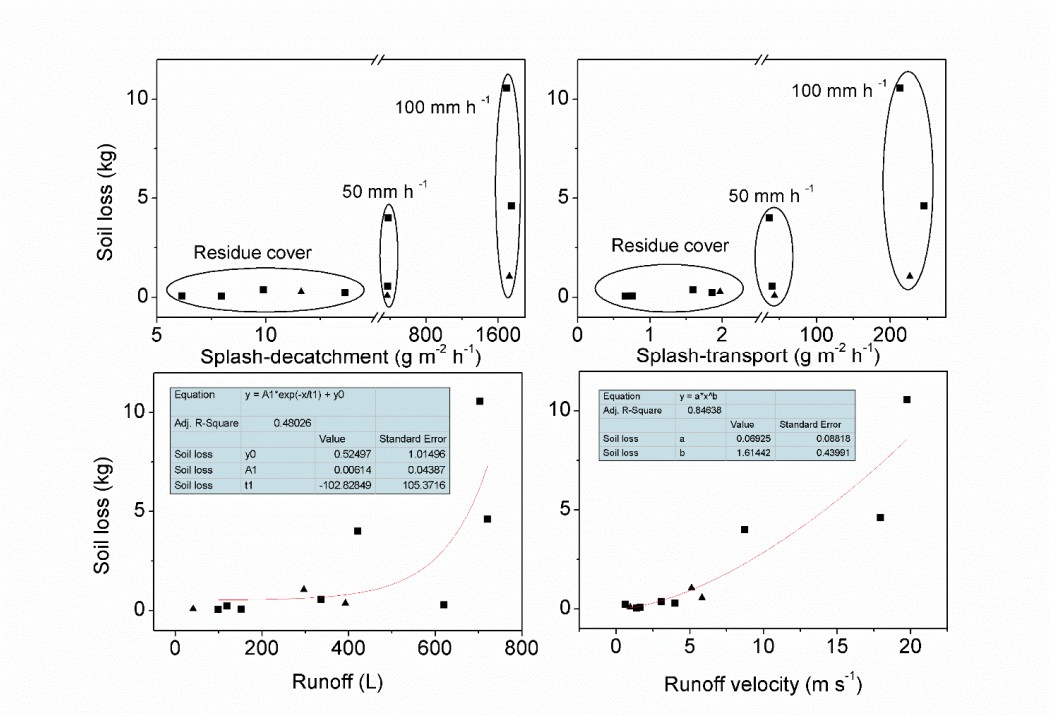

