# Peer review of "Horizontal ridging with mulching as the optimal tillage practice to reduce surface"

_EGUsphere, 2022_

## Author Comment (AC1)

**Reply to comments of RC1: 'Comment on egusphere-2022-1526', Pedro Batista, 01 Feb 2023**

**Dear reviewer Dr. Pedro Batista,**

We are very grateful to your constructive and helpful work and suggestions for our manuscript, egusphere-2022-1526. We are sure that your comments will greatly improve the quality of our manuscript. According with your further advices, we amended the relevant parts following your comments exactly. Revised portion are marked also in red in the revision this time. Comments were responded below one by one. We hope our revision would meet your request.

Thank you!

**general comments**

The manuscript describes the influence of different tillage practices on surface runoff and soil erosion in Mollisol maize plots, based on rainfall simulation experiments in Northeast China. Although I see the value in the research topic, I do not think this manuscript is ready to be considered for publication in SOIL. There is simply not enough information in the methodology to allow for a proper assessment of results. There are also multiple inconsistencies which, in my opinion, compromise the scientific quality of the manuscript.

For instance, the authors state that their first objective is to "identify influence of maize seedling canopy on soil loss". However, canopy cover was apparently not measured by the authors, or at least this was not reported. Moreover, although the manuscript seems to focus on crop seedling stages, there is no information regarding the timing of the rainfall simulations in relation to the crop stage. That is, the date(s) of the rainfall simulation(s) is(are) not provided, not even the number of days after sowing. There is also no information about the number of rainfall simulations performed per treatment, nor the number of plots per treatment. Hence, I do not know what the treatment means and error bars refer to in figures 4 and 5. This compromises the interpretation of the statistical analysis presented by the authors.

Furthermore, the authors report data on droplet size and kinetic energy for the rainfall simulations, but there is no information about how this data were measured. Besides the missing information, some of the methods seem unusual or lack justification (see detailed comments below regarding the "pre-rain" 24 hours before the experiments and the "drying of the topsoil layer"). In addition, I found some of the information presented in the introduction to be somewhat imprecise or not sufficiently supported by references.

A: Thank you for your professional and constructive comments and suggestion. You are right that the information of you point out were lack. In our revision, we replenished detail information about the time of simulation rainfall after maize seed sowed, the number of rainfall simulations performed per treatment, the measurement of droplet size and kinetic energy, pre-rain 24 hrs, and drying of the topsoil layer.

These and several other issues are listed in the detailed comments below.

**Detailed comments**

Line 44: Please consider changing "soil layer thinning" to "soil thinning".

A: Thank you for your professional suggestion. We changed it followed your suggestion as shown in line 43 in page 3.

Line 45: I suggest being more nuanced about crop yield losses associated to soil erosion (e.g., "and potentially to yield losses").

A: Thank you for your suggestion. We changed 'crop yield decline' to 'crop yield decline' in line 44 in page 3.

Lines 46-48: Is the statement "Mollisol regions […] are the major crop producing areas globally" accurate? I could not find the reference you provided (i.e., Zheng, 2020).

A: Yes, it is. The reference of Zheng edited in Chinese, and the detail information as shown in Lines 713-714 in page 24.

Lines 50-51: What is total soil loss area?

A: Thank you for your suggestion. We replenished the data of the total loss area in lines 49-50 in page 3.

Line 51: "Addressing soil erosion is important for soil loss reduction" seems redundant, please consider rephrasing.

A:Thank you for your suggestion. We replaced 'soil erosion' by 'slope erosion' in line 51 in page 3.

Lines 58-60: These statements sound strange to me (perhaps I misunderstood something). As far as I know, a very substantial amount of research has investigated interactions between vegetation and soil erosion, including at early crop development stages.

A: Thank you for your comment. Yes, there are large amounts of research on interactions between vegetation and soil erosion in the world, but we did not find a very substantial amount of research on soil erosion at early crop development stages from previous literature.

Line 62: Sorry, which region?

A: The region is the Northeast China as we mentioned in line 49. And, we also supplied the information in line 61 in page 3.

Lines 61-63: I had a hard time understanding this. Are the rainfall simulation studies related to the ones during the rainy season? Also, which rainy season? For which region?

A: Thank you for your comment. Yes, the rainfall simulation studies related to the ones during the rainy season as we cited the four references of Li et al., 2016; Liu et al., 2011; Lu et al., 2016; Xu et al., 2018. The rainy season is from July to September in the northeast China as we described in line 62 in page 3. We supplied the detail information of the region in line 61 in page 3.

Line 67: Do you mean soil water holding capacity? How is Figure 1 illustrating this statement?

A: Thank you for your comment. We deleted poor soil holding capacity because figure 1 can not illustrated this statement of soil holding capacity.

Lines 76-81: Please consider rewriting this paragraph.

A: Thank you. We rewrote this paragraph as shown in lines 76-81 in page 4.

Line 94: Could you please revise this sentence? By reading this I would understand the total soil depth is 30 cm, but I reckon this is not the case.

A: Thank you. We rewrote this sentence as shown in lines 94-95 in page 5.

Line 102: I am not familiar with the term "agglomerate impurities" in this context. Could you please explain/reformulate?

A: Thank you. We changed 'agglomerate impurities' to 'impurities' in 101 in page 5.

Lines 103-106: Sorry, but I did not understand this part of the methods. Could you reformulate?

A: Thank you. We rewrote this sentence as shown in lines 102-103 in page 5.

Line 107: Variety or cultivar?

A: Thank you for your professional comment. We changed 'variety' to 'cultivar' in line 104 in page 5.

Line 110: The "Flat-planting plots" had not been mentioned in the text yet, so I do not know what you are referring to here.

A: Thank you. We move them to lines 115-118 in page 6.

Line 122: How many plots?

A: There are 6 treatments and 18 plots. We replenished the detail information in lines 119-120 in page 6.

Line 125: Are you sure that one hour of rainfall with 100 mm $hr^{-1}$ intensity is representative of rainfall patterns in your study area?

A: Yes, we are sure that one hour of rainfall with 100 mm $hr^{-1}$ intensity is representative of rainfall patterns in our study area as the two references Xu et al. (2018) and Wang et al. (2021a) reported.

Line 127: How many plots? When were the simulations performed? How many days after sowing? Do you have information on canopy cover and plant height?

A: There are 6 treatments and 18 plots. We replenished the detail information in lines 119-120 in page 6. All simulated rainfall experiments stared from July 19, 2013, after 40 days of sowing, and we added this information in lines 106-107 in page 5. It is a pity, we did not measure canopy cover and plant height of maize plants.

Lines 127-130: How does pre-rain at 30 mm $hr^{-1}$ for 5 min 24 hours before the experiments ensure consistent soil moisture?

A: The pre-rain duration of 5 min is our lab experience as reported in Zhang et al.(2009b).

Line 131: How did you dry the topsoil layer after the experiments? This sounds a bit odd, maybe I misunderstood something. Also, what do you mean by rainfall event? Do you mean the simulation? I am sorry, but I find your methods difficult to understand (description- and rationale-wise).

A: We meant that the dry topsoil layer is removing waterproof canvas ceiling and surround canvases after each rainfall, and then sun and wind would dry the plots, we added the information in line 130 in page 6. We replaced rainfall event to rainfall simulation as shown in line 129 in page 6.

Line 151: As far as I understand, splash erosion would start as soon as the rainfall simulation begins. Moreover, how many rainfall simulations did you perform for each treatment?

A: Thank you for your professional comments. Yes, we start splash erosion at the rainfall simulation begins and hold about 15 min, we supplied the information in lines 150-151 in page 7. We repeated three times for each treatment.

Line 153: I found the statistical analyses difficult to understand without information regarding the number of plots and the number of rainfall simulations per treatment. That is, what are the "treatment means" you refer to? Also, what are the treatments? That is, how did you account for the interactions between tillage type, ridging direction, and rainfall intensities?

A: Thank you for your professional comments. We replenished the information as you pointed as shown in lines 119-120 in page 6. And, we did not analyze the interactions between tillage type, ridging direction, and rainfall intensities in this manuscript.

Lines 160-161: This is the first time you mention the measurement of raindrop energy and size distribution. How did you measure these? Shouldn't this information be in the methods?

A: Yes, you are right. Thank you for your suggestion. We supplied the information of the measurement of raindrop energy and size distribution as shown in lines 153-160 in page 7.

Lines 163-166: I found this very confusing. Please consider reformulating.

A: we rewrote the sentence as shown in lines 171-175 in page 8.

Lines 168-169: Do you think antecedent soil moisture might influence the time to the beginning of runoff?

A: Yes, you are right. Thank you for your professional comments. Yes, the antecedent soil moisture can influence the time to the beginning of runoff. But we think the influence would be same in our experiment.

Line 173: 23.8-fold is difficult to understand, please consider giving the actual time to runoff for the CK treatment.

A: Thank you. We replaced fold to times as shown in lines 182-183 in page 8.

Lines 174-175: Are these times to runoff referring to which rainfall intensity?

A: Thank you. We mean that the two rainfall densities.

Lines 190-191: Did these low runoff amounts cause the rupture of the ridges? Is this correct?

A: Yes, you are right. Thank you. Yes, we think low runoff amounts is one of the reasons cause the rupture of the ridges, which means that more rainfall water infiltration in soil and cause soil saturation, and damage soil structure, then make ridges rupture.

Lines 197-198: Augmented the soil loss in comparison to CK?

A: Yes, all soil loss were compared with CK.

Furthermore, we corrected the wrong expression of control and replaced it by CK in figures 4-7 in revision, improved the resolution of figure8, and the all 5 figures were replaced by new version.

Thank you again for your care and patience, and your professional and constructive comments and suggestion.

Please do not hesitate to contact us if you have any more comments and suggestion.

Best wishes,

Dr. Prof. Yubin Zhang ybzhang@jlu.edu.cn

[revised manuscript text omitted]

**Measurement of raindrop energy and drop-size distribution**

The measurement of rainfall energy and rain drop-size distribution was using splash pan and followed the method as reported by Qin et al. (2014).

The energy calculation equation is showed in formula (1):

$E = \rho \pi d^3 V_m^2 / 12$     (1)

Where, E is the rainfall energy, J; $\rho$ is rainfall density, which was measure at each simulation rainfall, kg/m$^3$; $\pi$ is a constant, 3.14; $d$ is the raindrop size, which was measured using splash-pan at each simulation rainfall, m; $V_m$ is the raindrop velocity, m/s.

**Data analysis**

All data were analyzed for statistical significance of treatment effects by one-way analysis of variance (ANOVA) using SPSS 16.0 (SPSS Inc., Chicago, IL, USA). The least significant difference (LSD) at $p<0.05$ was used to compare the treatment means. Plots were drawn using Origin 9.0 (Origin

Lab Corporation, Northampton, MA, USA).

**Results**

**Raindrop energy and distribution above/below corn seedling canopy**

As shown in Tables 1 and 2, the energy and size distribution of raindrops were significantly different between above and below the canopy of seedling corn. Under the two rainfall intensities, the canopy mitigation of raindrop energy was observed more in conservation than conventional tillage measures. Compared to above canopy, the percentage of raindrops of below canopy with less than 2.5

[revised manuscript text omitted]

---

## Author Comment (AC2)

**Reply to comments of RC2: 'Comment on egusphere-2022-1526', Josef Krasa, 07 Feb 2023**

**Dear reviewer Dr. Josef Krasa,**

We are very grateful to your interesting, your constructive and helpful work and suggestions for our manuscript, egusphere-2022-1526. We are sure that your comments will greatly improve the quality of our manuscript. According with your further advices, we amended the relevant parts following your comments exactly. Revised portion are marked also in red in the revision this time. Comments were responded below one by one. We hope our revision would meet your request.

Thank you!

**I thank for the option** to learn about interesting experiments.

I like the experiments, I like the goals, I like the focus of the authors on many variables in the study setup and the aim to use measured data to "explain".

But without justifying the data was obtained "correctly" and with repetitions I cannot take the assumptions seriously and discuss on the results.

The experiment remains very poorly described, so the manuscript cannot be accepted in the present form without reworking the methods and results. In the entire manuscript there is completely unclear how was the setup concerning replicates. In several parts some repetitions are "pointed" (e.g. L131-133) but absolutely hiding the real approaches. There is no description on the statistics used to analyse the single experiments and produce presented values.

In many parts methods used to get the data in not explained (eg. drop sizes, distributions, KE, etc.).

Many graphs and tables are only vaguely described.

These and several other issues are listed in the detailed comments below.

A: Thank you for your professional and constructive comments and suggestion. Thank you for your interesting. You are right that the information of you point out were lack. In our revision, we replenished detail information about the time of simulation rainfall after maize seed sowed, the number of rainfall simulations performed per treatment, the measurement of droplet size and kinetic energy, pre-rain 24 hrs, and drying of the topsoil layer.

**Detailed comments:**

L 23 – was this investigated directly, or is it authors' assumption?

A: Thank you. It was directly investigated by our experiment.

L 29 – comparison to buffers strips have no justification

A: Thank you for your professional suggestion. We corrected this sentence as shown in lines 27-28.

L 63 – what is the relation of rainfall simulation experiment to rain seasonality concerning the results interpretation? I do not understand the statement.

A: Thank you for your comment. Yes, the rainfall simulation experiment only can be processed in rain season on filed plots in Northeast China, and the rainfall simulation studies related to the ones during the rainy season as we cited the four references of Li et al. (2016), Liu et al. (2011), Lu et al. (2016), and Xu et al. (2018). The rainy season is from July to September in the northeast China as we described in line 62 in revision.

L 93 – I assume from the figures it is a natural slope – so it was rather selected than set to? (even if I understand that the soil profile was created by added topsoil material) Or does not the Figure 1 refer to the experimental area? That is not clear from the figure 1, hence the Figure 2 looks like different area. Maybe to clarify the relevance of the figures for the experiment setup.

A: Thank you for your professional suggestion. Yes, you are right. Figure 1 is a natural slope scenario and is widely distributed in Northeast China. We just want readers know what is the field situation in our study region through Figure 1. Figure 2 is our experiment field plots, where is in Binxian county belongs to Northeast China.

L 131 – 133: How the plots with maize could be restored the way you describe without affecting the vegetation. How many replicates could be done on the plots with vegetation. Or was it reseeded and used after longer period for replicates? Or were the replicates realized on other plots nearby? (The figure 2 does not look like) The whole process of plot maintenance and results replicability is very unclear.

A: Thank you for your professional comment. We did not reseed or realize on other plots nearby after simulation rainfall. We use the same collected soil to fill the rill gully and pad it, and supply same amounts of the loss cornstalk by pre-rainfall, the method was followed by Polyakov and Nearing reported in 2003.

L 136 – From he top? (not topsoil) How long plot section was used to estimate runoff velocity? 1m? What it means after it became steady? Was that always in the same minute? Or before the end of experiment?

A: Thank you for your professional suggestion. Yes, you are right. It was from top of the slope and we changed it as shown in line 135. The long plot section was 1m used to estimate runoff velocity. Steady means runoff continuously occurred. We just measured runoff velocity from runoff steady at the same time by three people.

L 141-142: I do not understand the sentence. What is the "runoff loss" in the context?

L 143 – delete "runoff rate".

A: Thank you for your professional suggestion. We deleted runoff rate.

L 160 – 166: Where is the methodology section for these results? How the data was obtained? Missing techniques, setup, repetitions, durations, ….

A: Thank you for your professional comment. We replenished the information in lines 152-159 in revision.

L 292 – runoff initiation

A: Thank you for your professional suggestion. We corrected it in line 299.

L 298 Why then 100mm/h were in the study focus?
[Figure]

A: Thank you for your professional comment. The intensity 100mm/h of rainfall is the extreme precipitation in our study region

L 322-325 One of the only sections raising questions on replicates, otherwise ignored in the whole text.

A: Thank you for your professional suggestion. Yes, you are right. We deleted it in revision.

L 391-392: Contradictory to the statement in L 298

A: Thank you for your professional comment. We did not think they are contradictory; the reason is extremely storm may occur in Spring in the study region as we described in lines 120-123 in revision.

Concerning the discussion and conclusions, I did read it with interest, but before re-working the above sections, I do not want to rise my detailed comments here.

A: Thank you for your interesting. We replenished them in revision.

Figure 8: Y-axis: is it soil loss, or sediment yield as referred to in whole manuscript (kg).

A: Thank you for your professional suggestion. Yes, you are right.

What is the way (units, values) the residue cover is interpreted in the Figure 8 – that is totally unclear.

A: Thank you for your professional comment. We just analyzed the effect of residue cover on soil loss with the results of splash detachment and transport, and the unit was same as other treats. At the same time, we improved the resolution of figure 8 and update it in revision.

Figures 4-7: control, not contorl.

A: Thank you. Yes, you are right. We corrected it and replaced it by CK. Figures 4-7 were updated in revision.

Furthermore, we improved other unclearly section and update in revision.

Thank you again for your care, patience and interesting, and your professional and constructive comments and suggestion.

Please do not hesitate to contact us if you have any more comments and suggestion.

Best wishes,

Dr. Prof. Yubin Zhang ybzhang@jlu.edu.cn

[revised manuscript text omitted]

**Measurement of raindrop energy and drop-size distribution**

The measurement of rainfall energy and rain drop-size distribution was using splash pan and followed the method as reported by Qin et al. (2014).

The energy calculation equation is showed in formula (1):

$E = \rho \pi d^3 V_m^2 / 12$     (1)

Where, E is the rainfall energy, J; $\rho$ is rainfall density, which was measure at each simulation rainfall, kg/m³; $\pi$ is a constant, 3.14; $d$ is the raindrop size, which was measured using splash-pan at each simulation rainfall, m; $V_m$ is the raindrop velocity, m/s.

**Data analysis**

All data were analyzed for statistical significance of treatment effects by one-way analysis of variance (ANOVA) using SPSS 16.0 (SPSS Inc., Chicago, IL, USA). The least significant difference (LSD) at $p<0.05$ was used to compare the treatment means. Plots were drawn using Origin 9.0 (Origin

Lab Corporation, Northampton, MA, USA).

**Results**

**Raindrop energy and distribution above/below corn seedling canopy**

As shown in Tables 1 and 2, the energy and size distribution of raindrops were significantly different between above and below the canopy of seedling corn. Under the two rainfall intensities, the canopy mitigation of raindrop energy was observed more in conservation than conventional tillage measures. Compared to above canopy, the percentage of raindrops of below canopy with less than 2.5

[revised manuscript text omitted]